# Cell type-specific and time-dependent light exposure contribute to silencing in neurons expressing Channelrhodopsin-2

Alexander M Herman[1†], Longwen Huang[2†], Dona K Murphey[3], Isabella Garcia[1,4], Benjamin R Arenkiel[1,2,3,5]*

[1]Program in Developmental Biology, Baylor College of Medicine, Houston, United States; [2]Department of Neuroscience, Baylor College of Medicine, Houston, United States; [3]Department of Molecular and Human Genetics, Baylor College of Medicine, Houston, United States; [4]Medical Scientist Training Program, Baylor College of Medicine, Houston, United States; [5]Jan and Dan Duncan Neurological Research Institute at Texas Children's Hospital, Houston, United States

**Abstract** Channelrhodopsin-2 (ChR2) has quickly gained popularity as a powerful tool for eliciting genetically targeted neuronal activation. However, little has been reported on the response kinetics of optogenetic stimulation across different neuronal subtypes. With excess stimulation, neurons can be driven into depolarization block, a state where they cease to fire action potentials. Herein, we demonstrate that light-induced depolarization block in neurons expressing ChR2 poses experimental challenges for stable activation of specific cell types and may confound interpretation of experiments when 'activated' neurons are in fact being functionally silenced. We show both ex vivo and in vivo that certain neuronal subtypes targeted for ChR2 expression become increasingly susceptible to depolarization block as the duration of light pulses are increased. We find that interneuron populations have a greater susceptibility to this effect than principal excitatory neurons, which are more resistant to light-induced depolarization block. Our results highlight the need to empirically determine the photo-response properties of targeted neurons when using ChR2, particularly in studies designed to elicit complex circuit responses in vivo where neuronal activity will not be recorded simultaneous to light stimulation.

*For correspondence: arenkiel@bcm.edu

†These authors contributed equally to this work

Competing interests: The authors declare that no competing interests exist.

## Introduction

Optogenetics is a powerful emergent technology for investigating complex patterns of synaptic connectivity and circuit properties that underlie physiology and behavior. By selectively expressing and activating light-gated ion channels or light-driven ion pumps, optical control of neural circuitry can be accomplished with high spatial and temporal resolution (*Boyden et al., 2005*; *Li et al., 2005*; *Nagel et al., 2005*). To date, many variants of light-sensitive proteins are available for use towards the activation or inhibition of excitable cells (*Lin, 2011*; *Rein and Deussing, 2012*). The most commonly used variant, channelrhodopsin-2 (ChR2) has become increasingly valuable in studies that exploit cell type-specific excitation.

Isolated from the green algae, *Chlamydomonas reinhardtii*, ChR2 is a seven-transmembrane-domain, nonselective cation channel that is directly activated by blue light when bound with the chromophore all-*trans*-retinal (*Hegemann et al., 1991*; *Lawson et al., 1991*; *Takahashi et al., 1991*; *Nagel et al., 2003*). Absorption of photons by the chromophore induces conformational changes in the channel that allow for the passage of cations across the cell membrane in which it is expressed (*Boyden et al., 2005*). This direct photo-gating makes ChR2 useful for manipulating neuronal activity in vitro (*Boyden*

**eLife digest** The brain is a highly complex structure composed of trillions of interconnecting nerve cells. The pattern of connections between these cells gives rise to the various brain circuits that govern how the brain functions. Understanding how the brain is wired together is important for determining how 'faulty circuits' contribute to various neurological disorders.

New optogenetic technique tools allow neuroscientists to turn on specific neurons simply by shining light on them. These techniques involve genetically manipulating the organisms so that their neurons express proteins that are activated when they are exposed to light of a particular wavelength. However, it is important to understand the limitations of this approach—including the possibility that the light might actually turn off some neurons—when using it to study animal behavior.

Now, Herman, Huang et al. show that shining light pulses for long durations onto neurons expressing a light-activated protein called channelrhodopsin-2 causes the neurons to become silenced rather than activated. Moreover, certain types of neurons, called interneurons, are more susceptible to this effect—termed 'depolarization block'—than the other types of neurons.

Researchers need to be mindful of this effect when channelrhodopsin-2 is used in optogenetic experiments to study the behavior of living animals. However, this silencing property could be useful in experiments that investigate situations in which depolarization block is thought to contribute to brain function and health: such as in the treatments of schizophrenia and Parkinson's disease.

---

*et al., 2005*; *Li et al., 2005*; *Zhang and Oertner, 2007*) and in vivo (*Markram et al., 2004*; *Li et al., 2005*; *Nagel et al., 2005*; *Petreanu et al., 2007*; *Douglass et al., 2008*; *Huber et al., 2008*; *Mahoney et al., 2008*; *Ayling et al., 2009*; *Baier and Scott, 2009*; *Guo et al., 2009*; *Liu et al., 2009*; *Zhu et al., 2009*; *Katzel et al., 2011*; *Schultheis et al., 2011*; *Boyd et al., 2012*; *Britt et al., 2012*; *English et al., 2012*; *Beltramo et al., 2013*; *Chen et al., 2013*; *Chiu et al., 2013*; *Owen et al., 2013*). Coupled with genetically-targeted expression, ChR2 affords exquisite functional control of specific neuronal subpopulations. Many studies to date that utilized ChR2 for in vivo manipulation of behavior and/or circuit function have largely focused on the activation of principal excitatory neurons, particularly in the cortex (*Petreanu et al., 2007*; *Huber et al., 2008*; *Ayling et al., 2009*; *Boyd et al., 2012*; *Britt et al., 2012*; *Beltramo et al., 2013*; *Chen et al., 2013*). As the experimental applications of ChR2 move to include biophysically diverse interneurons (*Markram et al., 2004*; *Katzel et al., 2011*; *Schultheis et al., 2011*; *English et al., 2012*; *Chiu et al., 2013*; *Owen et al., 2013*), a fuller understanding of its possibilities and limitations becomes essential.

Although ChR2 expression, trafficking, and activation has been achieved in most types of nervous tissue (*Li et al., 2005*; *Nagel et al., 2005*; *Bi et al., 2006*; *Schroll et al., 2006*; *Adamantidis et al., 2007*; *Zhang et al., 2007*; *Douglass et al., 2008*; *Mahoney et al., 2008*; *Baier and Scott, 2009*; *Guo et al., 2009*; *Han et al., 2009*; *Liu et al., 2009*; *Zhu et al., 2009*; *Gourine et al., 2010*; *Hagglund et al., 2010*; *Diester and et al, 2011*; *Figueiredo et al., 2011*; *Sasaki et al., 2012*; *Ljaschenko et al., 2013*), little consideration has been given to the kinetics that constrain light-induced firing properties in different neuronal subtypes. Factors that affect the degree of neuronal photostimulation, including intrinsic differences in firing dynamics, membrane properties, and channel composition differ among neuronal cell types. These properties render certain neuronal populations more susceptible to depolarization block, which is generally induced by excessive cation influx, prolonged membrane depolarization, and/or the inability to repolarize, resulting in the failure to support continuous firing of action potentials. In vitro electrophysiological recordings have shown the ability of ultrafast microbial opsins to inactivate neurons via excessive photo-induced currents, where regular-spiking excitatory neurons showed a greater tendency to enter depolarization block than fast-spiking interneurons (*Mattis et al., 2012*). Similarly, recent studies in primary hippocampal cultures have explored the effect of different ChR2 variants on neuronal firing properties in response to differing light intensity and pulse duration (*Lin et al., 2013*). Such reports characterizing channelrhodopsin behavior highlight the need for empirically determining optimal light stimulation parameters for translation in vivo, as many factors including pulse duration, light intensity, ChR2 variant, method of expression, and targeted cell type may all affect a cell's response to light activation.

To date, little attention has been given to the extent and specificity of depolarization block subsequent to ChR2-mediated excitation. Using multiple ChR2 mouse lines that expressed the common ChR2 variant (H134R), we show both ex vivo and in vivo that light-induced neuronal firing can be accompanied and/or followed with periods of latency in various neuronal subtypes. Further, we show that interneurons are particularly vulnerable to this effect. In particular, at a fixed moderate frequency of light stimulation, increasing pulse width resulted in transient activation and protracted silencing of regular-spiking interneurons, but excitatory principal/pyramidal cells or subsets of fast-spiking neuronal populations were more resistant to depolarization block. Our data highlight that especially for interneurons, it is important to empirically define stimulation parameters to elicit desired effects. In particular, careful considerations must be taken when applying in vivo optogenetic approaches in animal studies, being mindful of the time intervals chosen for light pulses used to activate neurons as temporal changes in the photo stimulus duration may result in variable or unpredictable response patterns. Additionally, we found that the duration of a light pulse beyond a certain point, independent of frequency, fails to result in higher spike rates in neurons expressing ChR2, but rather silences them. Together, our data recommend the use of short-width light pulses when using ChR2 for the activation of interneuron cell types, while slightly longer pulse widths may be desirable for consistent spiking of larger excitatory cell types. Finally, our findings also support the possibility of exploiting this effect to investigate the physiological phenomena that underlie depolarization block, or targeted neuronal silencing.

## Results

### Extended light pulse duration increases susceptibility for depolarization block in interneurons

To date, little consideration has been given to the kinetics that constrain light-induced firing properties in different neuronal subtypes. To test for differential effects of ChR2-mediated light stimulation between different classes of neurons, we first performed whole cell electrophysiological recordings in acute brain slices from several classes of commonly studied interneurons and principal excitatory cells that express the ChR2-EYFP fusion protein.

The first type of interneuron we investigated has been characterized by its expression of corticotropin-releasing hormone (CRH) (*Taniguchi et al., 2011*). CRH serves diverse physiological roles in the nervous system as a hormonal regulator of stress (*Bale and Vale, 2004*), feeding (*Richard et al., 2000*), and reproduction (*Laatikainen, 1991*), among others, and is highly expressed in neurons of the hypothalamus (*Cusulin et al., 2013*), interneurons of the olfactory bulb (*Huang et al., 2013*), and sparsely throughout the cortex. To investigate the photo response properties of these cells, we crossed male *Crh-Cre*[+/−] animals with conditional female *ROSA26-lox-stop-lox-ChR2-EYFP* animals (Gt(ROSA)26Sor[tm32.1(CAG-COP4*H134R/EYFP)Hze/J], herein referred to as ROSA26[LSL-ChR2-EYFP]). Within the main olfactory bulb, *Crh-Cre*[+/−]; *ROSA26*[LSL-ChR2-EYFP] animals expressed the ChR2-EYFP fusion protein selectively in CRH-positive interneurons of the external plexiform layer (EPL) (*Figure 1A*). To investigate the photo-response kinetics of these cells, we performed whole-cell recordings from CRH-expressing EPL interneurons (N = 10 cells) to graded light-on/light-off stimulation parameters (*Figure 1B*). For each parameter, we used a single stimulation train consisting of 20 light pulses at 20 Hz. Initial stimulation parameters varied only with respect to 'time on' pulse width so as to determine the effect of light pulse duration on ChR2-mediated neuronal excitation, with fixed light intensity and frequency, which was kept at ≈40 mW/mm$^2$ and 20 Hz, respectively. For olfactory bulb CRH interneurons, we observed that short duration (5–25 ms) light pulses ('time on') were optimal for sustaining consistent neuronal firing, with a highest average firing rate elicited at 10 ms pulse widths (*Figure 1C*). In contrast, increasing the duration of a light pulse to more than 25 ms, such that 'time on' pulse duration outlasted the interval between pulses ('time off'), reliably decreased firing rates, and completely silenced neurons at pulse durations of ≥49 ms 'time on' (*Figure 1D*). This phenomenon is not likely due to desensitization of the ChR2 itself, as stepwise current injections independent of ChR2-mediated light activation mirror silencing observed in neurons from increased light pulse duration (*Figure 1E,F*). Interestingly, CRH-expressing EPL interneurons exhibit high probability (>80%) co-expression with the fast-spiking interneuron marker, parvalbumin (PV), and are capable of achieving high firing rates when stimulated by light (*Huang et al., 2013*). Our present results suggest that CRH-expressing EPL interneurons exhibit a functional heterogeneity in their electrical responses, at least in a subset of neurons, highlighting the limitation of using broad neuronal markers (e.g., parvalbumin) as being indicative of a general firing

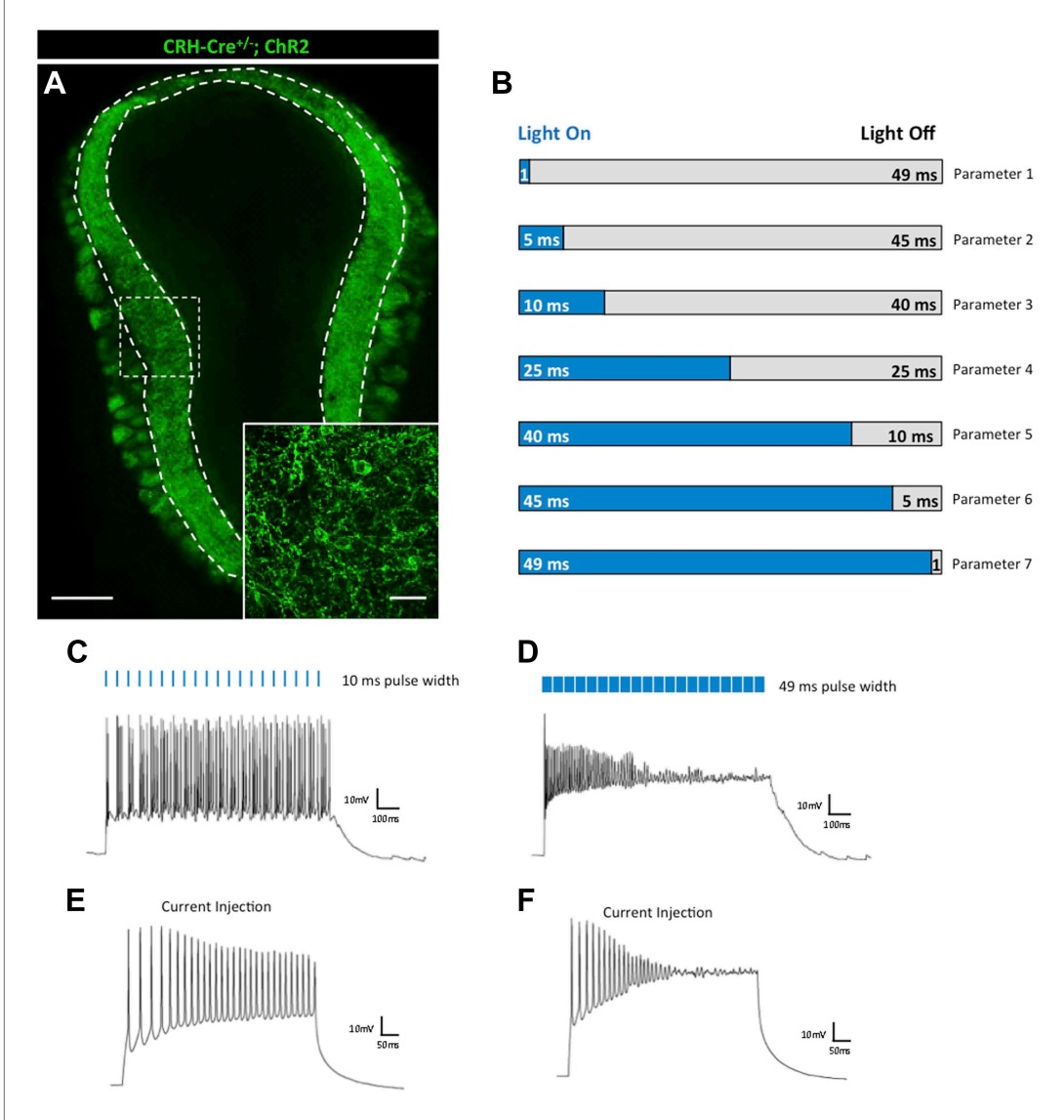

**Figure 1**. Effects of light pulse duration on CRH-expressing interneurons of the olfactory bulb. (**A**) *Crh-Cre$^{+/-}$; ROSA$^{LSL-ChR2-EYFP}$* mice express the ChR2-EYFP fusion protein in CRH-expressing interneurons of the external plexiform layer (EPL) of the main olfactory bulb (scale bar, 0.5 mm). Inlay represents zoomed image of ChR2-expressing interneurons of the EPL (scale bar, 100 μM). (**B**) Firing responses of ChR2-expressing neurons were recorded for seven different stimulation parameters. Each light stimulation parameter consists of a single train comprised of 20 light pulses (≈40 mW/mm²) at 20 Hz. Pulse width is the only condition that varies among the seven parameters. Parameter 1–1 ms pulse width/49 ms intervals, Parameter 2–5 ms pulse width/45 ms intervals, Parameter 3–10 ms pulse width/40 ms intervals, Parameter 4–25 ms pulse width/25 ms intervals, Parameter 5–40 ms pulse width/10 ms intervals, Parameter 6–45 ms pulse width/5 ms intervals, Parameter 7–49 ms pulse width/1 ms intervals. (**C**) Robust firing of a CRH-expressing EPL interneuron in response to brief light pulses (20 Hz, 10 ms pulse width). (**D**) Prolonged light pulse duration (20 Hz, 49 ms pulse width) leads to depolarization block in CRH interneurons. (**E**) Moderate current injection (60 pA) elicits regular firing of ChR2-expressing CRH interneurons. (**F**) High current injection (160 pA) results in depolarization block of ChR2-expressing CRH interneurons.

response pattern (i.e., fast-spiking) that may not be generalizable from one brain region (e.g., cortex) to another (e.g., olfactory bulb).

To further characterize this phenomenon across other cell types, we next performed whole-cell electrophysiological recordings from two types of choline acetyltransferase (ChAT)-expressing interneurons in a mouse line that expressed ChR2-EYFP under the control of the ChAT promoter (*Zhao et al., 2011*). Cholinergic signaling has been implicated in reward, learning, and addiction (*Tapper et al., 2004*; *Maskos et al., 2005*; *De Biasi and Dani, 2011*), and optogenetic tools are now being broadly applied

to characterize the neural mechanisms underlying these diverse behaviors (*Witten et al., 2010*; *Ren et al., 2011*). *Chat-ChR2* transgenic mice show expression of ChR2-EYFP in cholinergic neurons of the brain, with dense expression in the medial habenula (MHB) (*Figure 2A*) and throughout the striatum

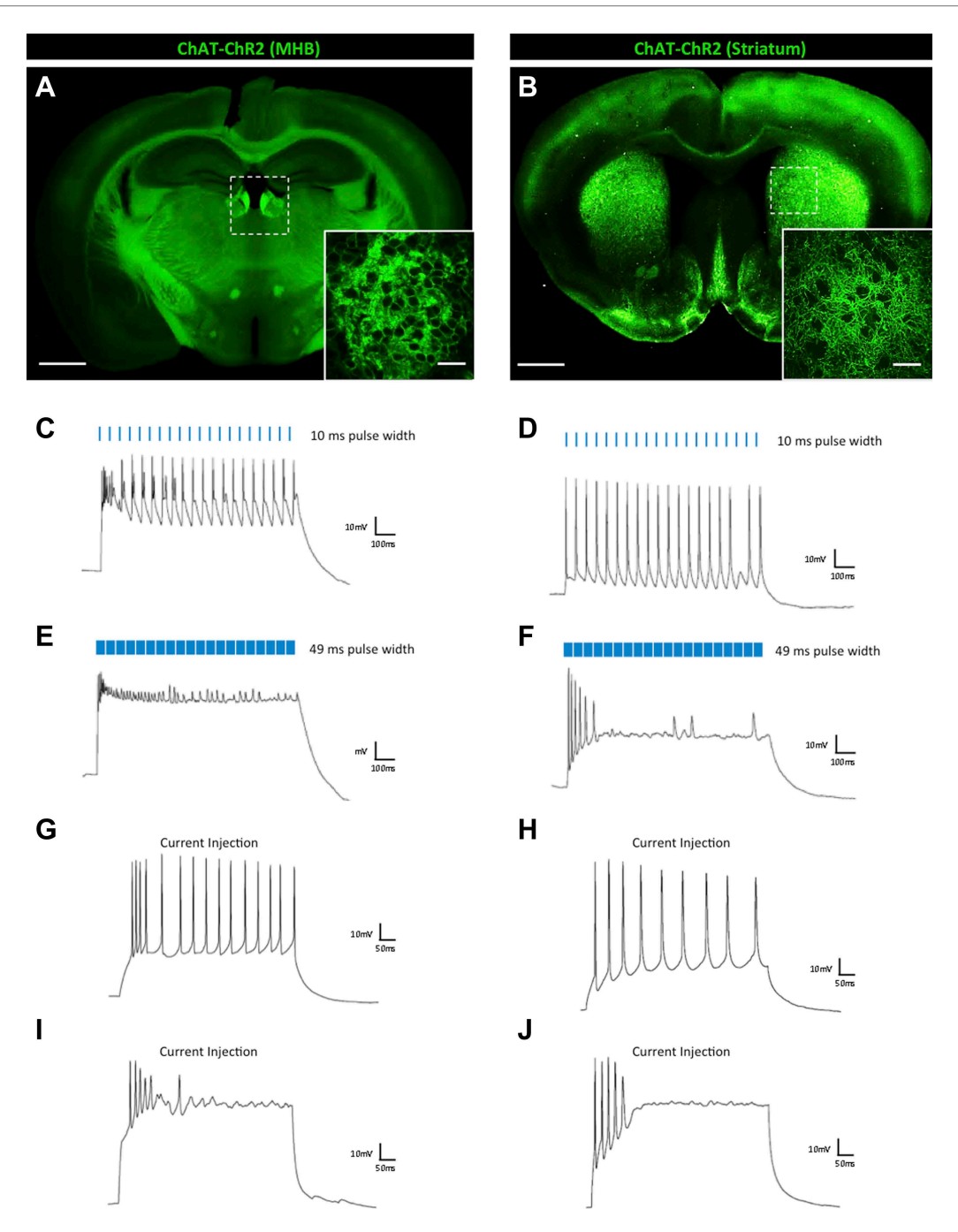

**Figure 2**. Effects of light pulse duration on ChAT-expressing interneurons of the MHB and striatum. (**A**) *Chat-ChR2* transgenic mice display ChR2-EYFP expression in the medial habenula (MHB) (scale bar, 1 mm) and (**B**) striatum (scale bar, 1 mm). Inlays show zoomed images of respective ChR2-expressing MHB or striatal interneurons (scale bars, 100 μM). Regular firing of ChR2-expressing ChAT interneurons of the (**C**) MHB and (**D**) striatum in response to brief light pulses (20 Hz, 10 ms pulse width). Prolonged light pulse duration (20 Hz, 49 ms pulse width) leads to depolarization block in (**E**) MHB and (**F**) striatal ChAT interneurons. Moderate current injection results in steady firing of ChR2-expressing (**G**) MHB (50 pA) and (**H**) striatal (80 pA) ChAT interneurons. High current injection results in depolarization block of ChR2-expressing (**I**) MHB (150 pA) and (**J**) striatal (300 pA) ChAT interneurons.

(*Figure 2B*). Like CRH-expressing interneurons of the olfactory bulb, ChAT-positive MHB interneurons (*Figure 2C*, N = 9) and striatal interneurons (*Figure 2D*, N = 15) exhibited robust and consistent firing with short duration (≤25 ms) light pulses, and dramatic decreases in firing rates with pulse width durations greater than 40 ms (*Figure 2E,F*). Again, this effect recapitulates the firing response to step-wise current injections (*Figure 2G–J*).

While the photocurrent responses of ChAT-expressing interneurons were largely uniform and comparable, somatostatin (SST)-positive interneurons demonstrated greater diversity of responses. SST-expressing interneurons have been characterized as a heterogeneous population, both morpho-logically and electrophysiologically (*Markram et al., 2004*; *Wang et al., 2004*; *Ma et al., 2006*). Distinct morphologies could be observed of SST interneurons throughout all layers of the cortex. Whereas those that resembled Martinotti cells made up a subset of the population (*Figure 3A,B*, white arrows), others resembled multipolar basket cells with varicose axonal projections (*Figure 3C,D*, open arrows), or appeared as small ovoid bitufted/bipolar cells (*Figure 3E*). To investigate the effects of light on these cells, we crossed male *Sst-Cre*$^{+/-}$ animals with floxed conditional female ROSA26$^{LSL-ChR2-EYFP}$ animals to target ChR2-EYFP expression to SST-positive interneurons throughout the cerebral cortex (*Figure 3F*). Our recordings validate previous reports (*Wang et al., 2004*; *Ma et al., 2006*) of elec-trophysiological heterogeneity between SST cortical interneurons, with one subset of SST interneurons exhibiting regular-spiking neuronal activity and another exhibiting faster-spiking dynamics. The former subset showed photostimulation responses consistent with those seen in CRH and ChAT interneurons (*Figure 3G–J*, N = 9). Interestingly, faster-spiking SST interneurons (N = 9) increased their firing rates with longer light pulse duration (*Figure 3K,L*). Furthermore, current injections (*Figure 3M,N*) recapitulated this result. This type of heterogeneity, even within a subclass of interneuron characterized by subtype-specific patterns of gene expression, urges further caution when designing light stimulation parame-ters so as to avoid potential effects of light-induced depolarization block, and to optimize for consistent and reliable neuronal firing among targeted populations of neurons.

Because light intensity may also affect ChR2-mediated spiking, we recorded firing dynamics in response to increasing pulse widths with different light intensities (≈10, 20, 30, and 40 mW/mm$^2$). Though overall average firing rates scaled with increasing light intensities across all interneuron cell types tested, increased pulse duration was sufficient to drive neurons into depolarization block regardless of light intensity (*Figure 4A–D*). Fast-spiking SST interneurons, however, exhibited increased and sustained firing with increasing pulse duration at all four light intensities (*Figure 4E*). These data are consistent with decreased averaged spike amplitudes over the duration of the stimulus train in response to increased pulse widths (*Figure 4F–I*), with the exception of fast-spiking SST interneurons, which showed only modest reductions in spike amplitudes over time (*Figure 4J*). While interneurons generally exhibited high spike probabilities with short light pulses (≤25 ms), prolonged light pulses dramatically increased spike failure (*Figure 4K*). With long pulses, regular-spiking interneurons characteristically fired an initial action potential or a short burst of action potentials, followed by prolonged depolarization block. Fast-spiking SST interneurons exhibited higher spike fidelity than other interneurons tested, even with long pulses (*Figure 4K*).

## Principal cell types are refractory to light-induced depolarization block

Due to size, typical excitatory neurons have low membrane resistance. This property makes excitatory neurons more resistant to depolarization block, as changes in voltage across the membrane require higher levels of current input. However, membrane channel composition promotes regular-spiking activity in these neurons, and thus likely establishes innate firing limits. Previous work has shown that increasing the frequency of light stimulation by various activating opsins results in decreased spike probability (*Mattis et al., 2012*), consistent with known firing properties of pyramidal neurons and fast-spiking PV interneurons. To test the response kinetics of light-induced firing in excitatory cell types, we next performed whole-cell recordings in brain slices from *Thy1-ChR2* mice that selectively express ChR2-EYFP in mitral/tufted cells of the main olfactory bulb (*Arenkiel et al., 2007*) (*Figure 5A*), as well as in layer V cortical pyramidal neurons (*Figure 5B*). In contrast to the observed inhibition reported in regular-spiking interneurons, these populations of excitatory cells were more resistant to prolonged light pulse-induced depolarization block. Increased light pulse duration elicited rapid, sustained neuronal firing in mitral cells (N = 14) (*Figure 5C,D*), whereas short pulse widths (<25 ms) appeared less effective than slightly longer pulse durations (25–49 ms) at eliciting robust spiking, likely due to low membrane resistance. This effect appeared to plateau at light pulse durations ≥40 ms, even with 5 s continuous

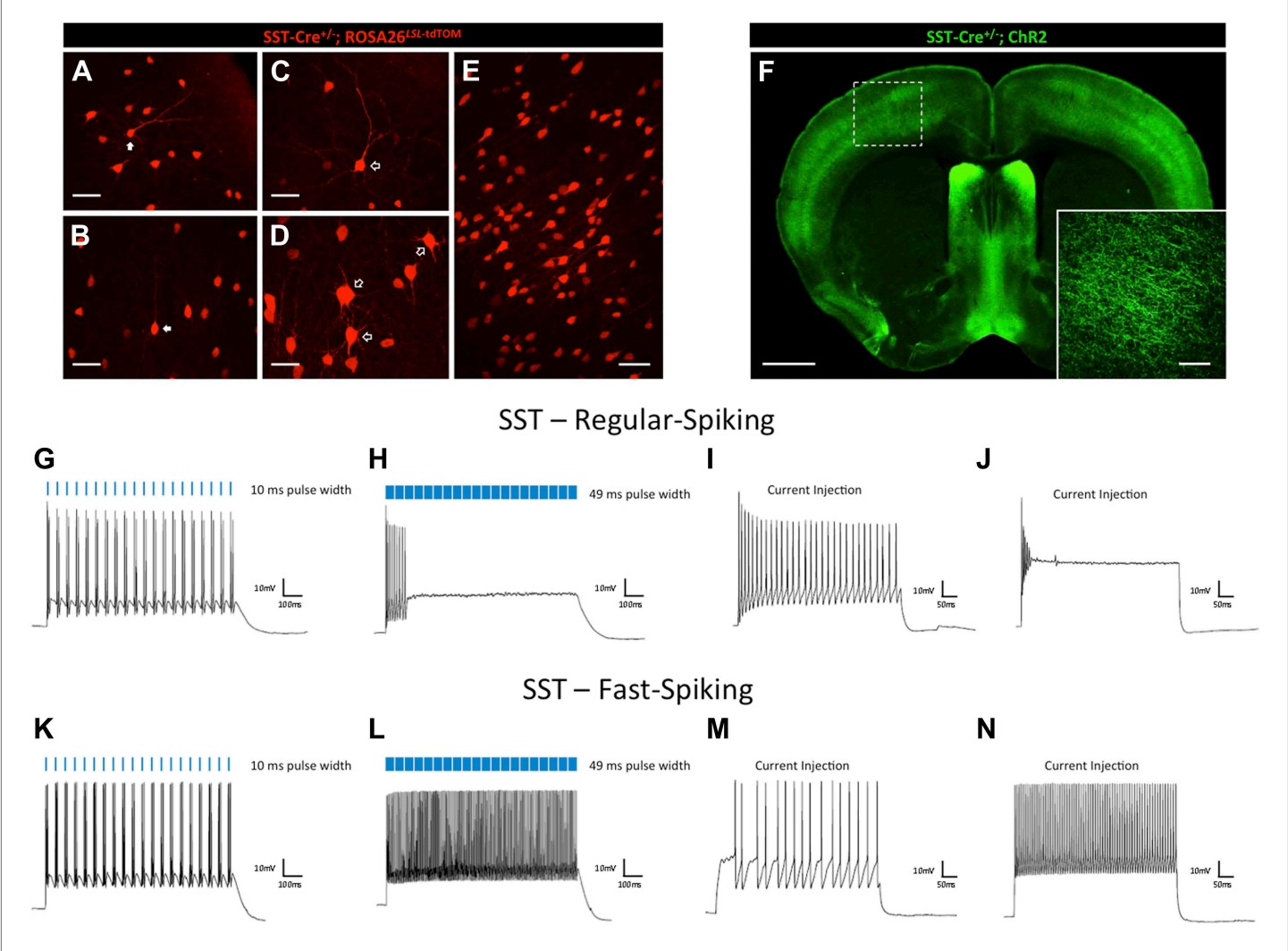

**Figure 3**. Effects of light pulse duration on a heterogeneous SST interneuron population. (**A–E**) *Sst-Cre^{+/−}; ROSA26^{LSL-tdTomato}* animals express the tdTomato reporter in SST-positive interneurons, which display heterogeneous morphologies (scale bars, 50 µM). (**F**) *Sst-Cre^{+/−}; ROSA^{LSL-ChR2-EYFP}* mice display diffuse expression of ChR2-EYFP throughout the cortex (scale bar, 1 mm). Inlay shows zoomed image of ChR2-expressing SST cortical interneurons (scale bar, 100 µM). (**G**) Steady firing of a regular-spiking ChR2-expressing SST cortical interneuron in response to brief light pulses (20 Hz, 10 ms pulse width). (**H**) Prolonged light pulse duration (20 Hz, 49 ms pulse width) leads to depolarization block in regular-spiking SST cortical interneurons. (**I**) Moderate current injection (30 pA) leads to steady firing of regular-spiking SST cortical interneurons expressing ChR2. (**J**) High current injection (100 pA) results in depolarization block of regular-spiking ChR2-expressing SST cortical interneurons. (**K**) Steady firing of a fast-spiking SST cortical interneuron in response to brief light pulse stimulation (20 Hz, 10 ms pulse width). (**L**) Prolonged light pulse duration (20 Hz, 49 ms pulse width) leads to robust firing in fast-spiking SST cortical interneurons. (**M**) Current injection (120 pA) leads to steady firing of fast-spiking SST interneurons. (**N**) High current injection (500 pA) results in robust firing of fast-spiking SST cortical interneurons.

light pulse stimulation (data not shown), though at this duration, 1 of 5 cells tested exhibited depolarization block, while the remaining cells did not. Pyramidal cells were consistently driven at pulse widths up to 40 ms (***Figure 5E,F***), but were more susceptible to depolarization block than mitral cells at longer pulse durations. Though reduced light intensities attenuated average firing rates in mitral cells, consistent upward trends in firing were observed at all four light intensities, even at extended pulse durations (***Figure 6A***). Cortical pyramidal cells were more resistant to depolarization block than other interneurons tested, but also tended to display characteristic silencing beyond 40 ms (***Figure 6B***). Similar to fast-spiking SST interneurons, mitral cells displayed modest amplitude reductions over the duration of a stimulus train during prolonged light pulses (***Figure 6C***). Pyramidal cells exhibited slight amplitude reductions up to 40 ms pulse widths before drastic reductions were observed at longer

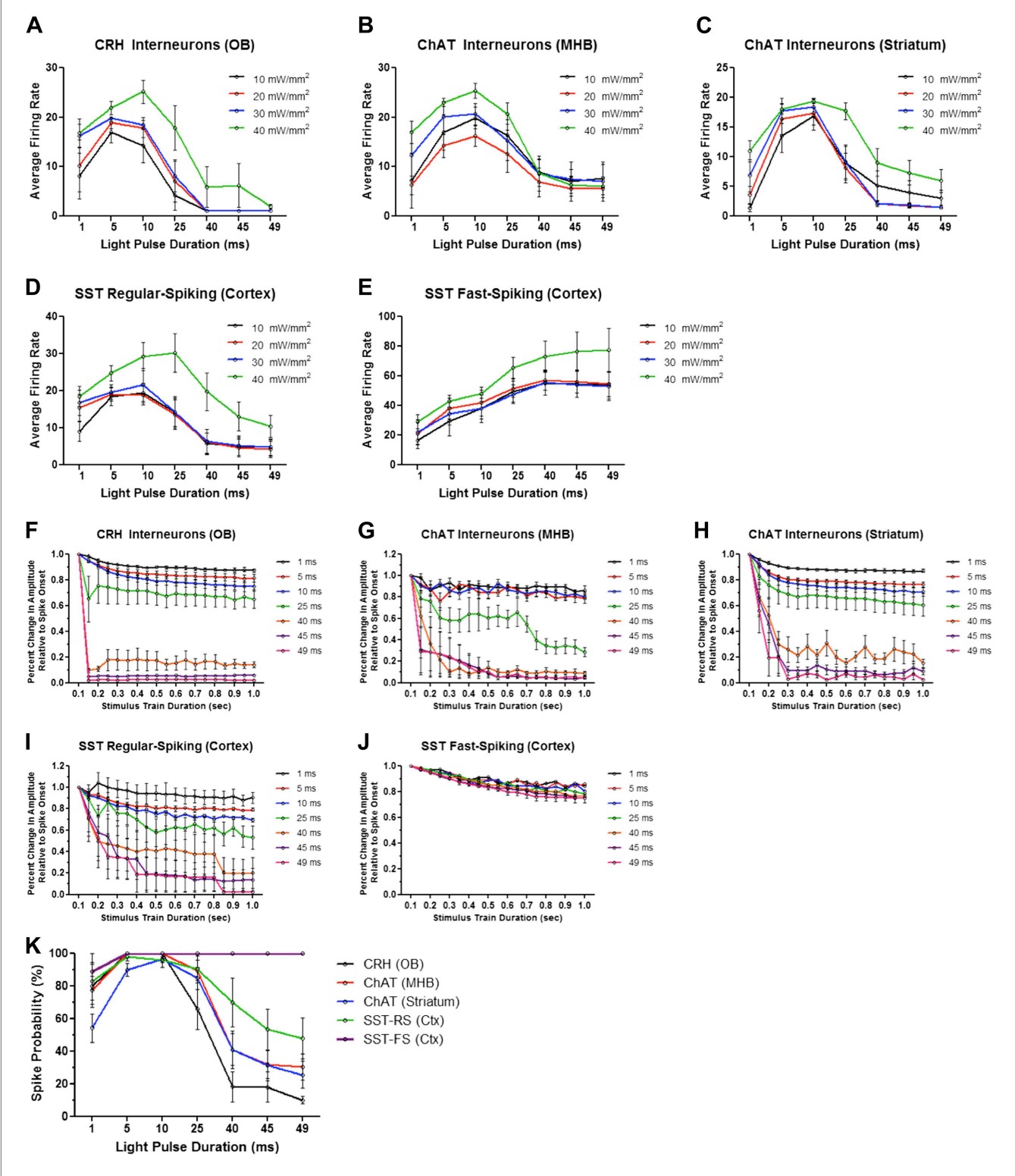

**Figure 4**. Firing dynamics of diverse interneurons in response to varying light pulse duration. Average firing rates of ChR2-expressing (**A**) CRH (N = 5–10 cells/intensity from 4 animals), (**B**) ChAT MHB (N = 4–9 cells/intensity from 5 animals), (**C**) ChAT striatal (N = 6–15 cells/intensity from 5 animals), (**D**) regular-spiking SST (N = 5–9 cells/intensity from 4 animals), and (**E**) fast-spiking SST interneurons (N = 4-9 cells/intensity from 4 animals) in response to variable light intensity and pulse duration (20 Hz). (**F–J**) Interneuron amplitudes, normalized to spike onset, in response to increasing pulse width. (**K**) Spike probabilities of various interneuron cell types in response to increasing pulse widths. Data points represent averages ± SEM. OB = Olfactory Bulb, MHB = Medial Habenula.

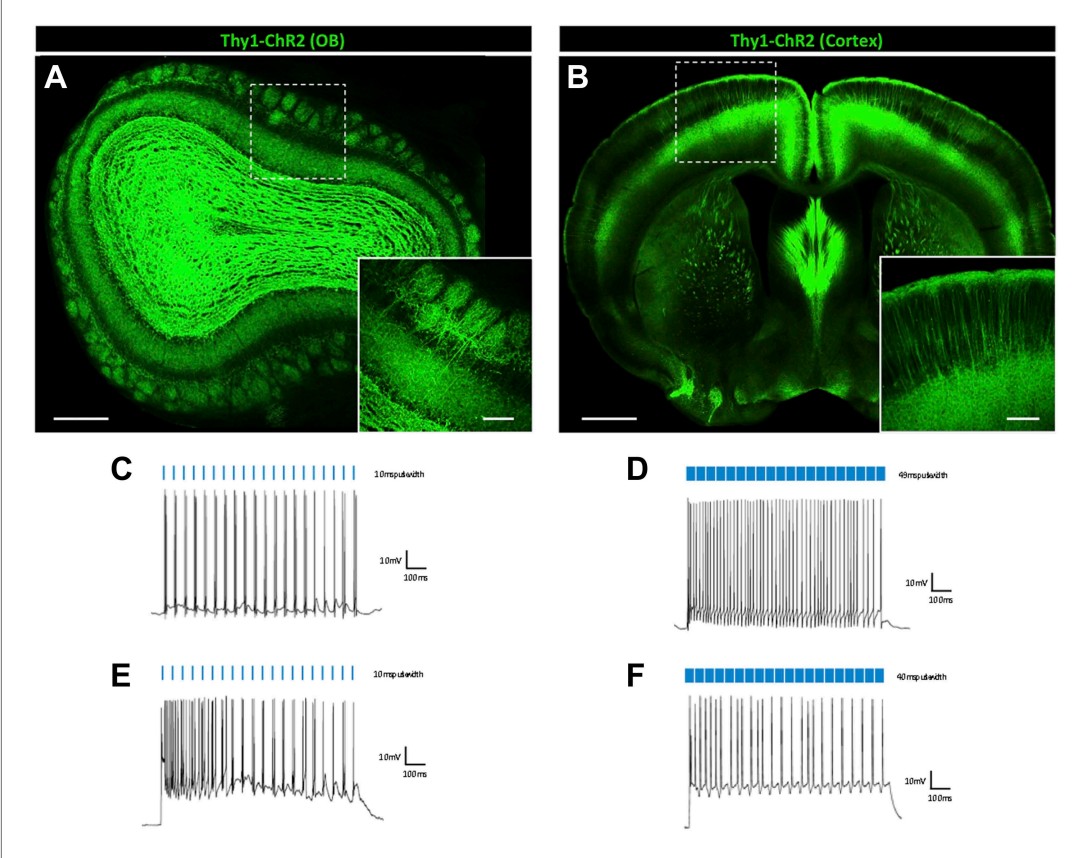

**Figure 5**. Principal excitatory cell types are less susceptible to light-induced depolarization block. *Thy1-ChR2* transgenic mice display ChR2-EYFP expression in (**A**) excitatory mitral cells of the main olfactory bulb (scale bar, 0.5 mm)—note that ChR2 expression is also observed throughout the granule cell layer of the olfactory bulb due to axon collaterals from mitral/tufted cells and ChR2-expressing centrifugal inputs from the piriform cortex–and (**B**) layer V cortical pyramidal neurons (scale bar, 1 mm). Inlays display zoomed images of ChR2-expressing olfactory bulb mitral cells (scale bar, 100 μM) or cortical pyramidal neurons (scale bar, 200 μM). (**C**) Mitral cells display steady firing in response to brief light pulses (20 Hz, 10 ms pulse width) and (**D**) enhanced firing in response to prolonged light pulse duration (20 Hz, 49 ms pulse width). (**E**) Steady firing of ChR2-expressing pyramidal cells in response to brief light pulse stimulation (20 Hz, 10 ms pulse width) and (**F**) prolonged light pulse duration (20 Hz, 40 ms pulse width). OB = Olfactory Bulb.

durations (*Figure 6D*). Interestingly, while brief light pulses were better at promoting higher spike fidelity in pyramidal cells, they were often not sufficient for eliciting robust firing in mitral cells, resulting in reduced spike probability (*Figure 6E*). Notably, extra spikes, particularly with longer pulses, were observed in all cell types. In fact, minimal pulse widths for all cell types tested were determined to have sub-millisecond response kinetics (*Figure 6F*), likely contributing to the observation of spike doublets and triplets when neurons were stimulated with long pulses. Together, these data show that different neuronal subtypes respond with drastically different response kinetics when stimulated with ChR2, and that the proper photostimulation parameters should be determined to elicit the desired firing output in target neurons.

## Cell type specificity and time-dependent light exposure contribute to depolarization block in vivo

The potential for unknowingly silencing neurons that express ChR2 is greatest when firing properties of targeted neurons are not recorded simultaneously with light exposure. This is often the case in optogenetic manipulations used to elicit behaviors in freely behaving animals. Our ex vivo data suggest that studies of ChR2-expressing interneurons may require empiric determination of the pulse duration that elicits the desired activation of neurons. To validate these findings in vivo, we next performed extracellular recordings in anesthetized animals that express ChR2 in SST-expressing cortical interneurons, olfactory bulb mitral cells, and cortical pyramidal neurons, and measured their firing activity in response to differing light pulse durations. Consistent with observations made in slices, regular-spiking SST

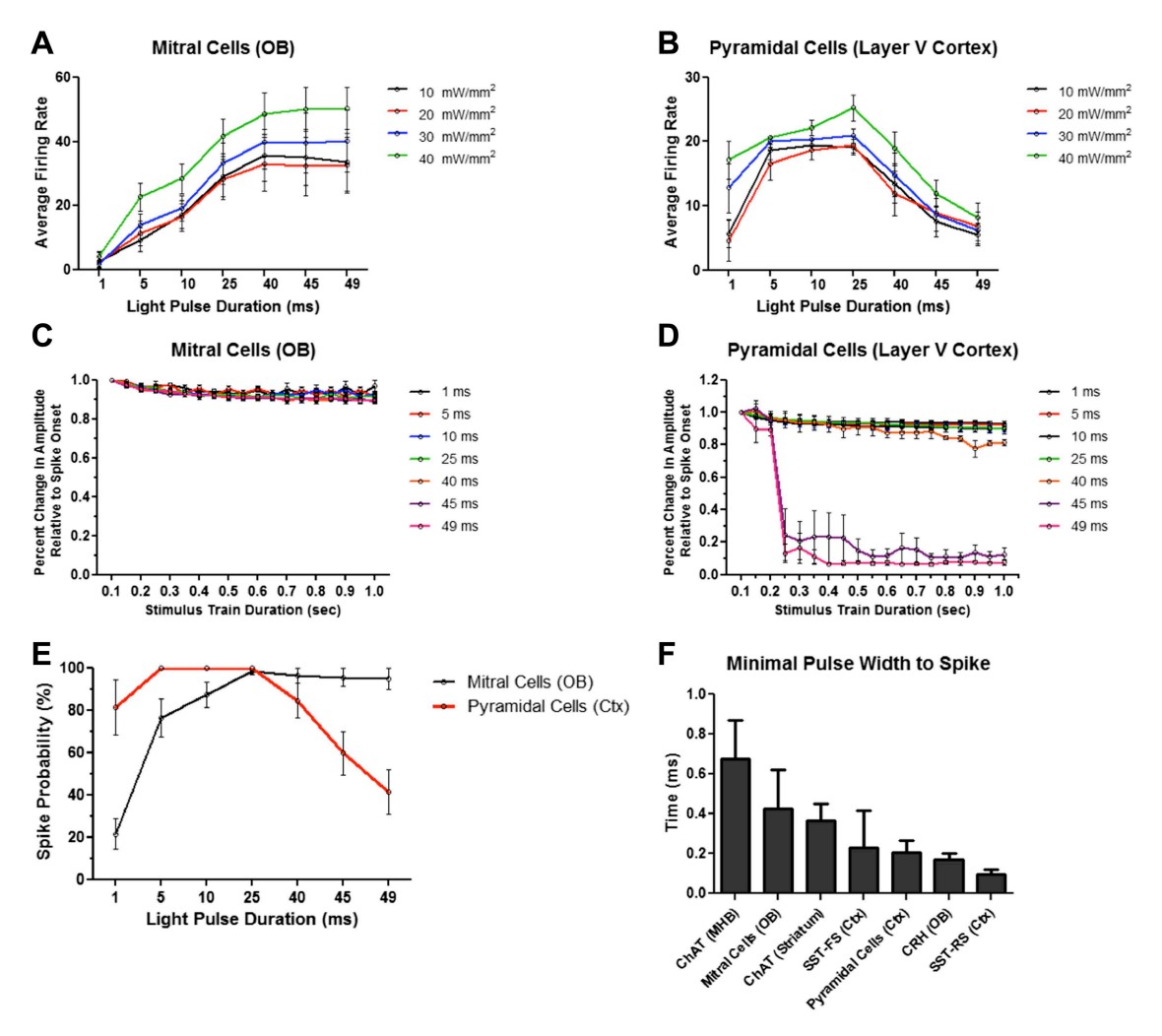

**Figure 6**. Firing dynamics of principal cell types in response to varying light pulse duration. Average firing rates of (**A**) mitral cells (N = 8–14 cells/intensity from 4 animals) and (**B**) cortical pyramidal cells (N = 6 cells/intensity from 3 animals) in response to variable light intensity and increasing pulse width (20 Hz). (**C**) Mitral cell and (**D**) pyramidal cell amplitudes, normalized to spike onset, in response to increasing pulse width. (**E**) Spike probabilities of mitral cells and cortical pyramidal neurons in response to increasing pulse widths. (**F**) Minimal pulse widths required to elicit single action potentials in various neuron populations. Data points represent averages ± SEM.

interneurons exhibited robust silencing due to prolonged light pulse duration (**Figure 7A**). Though in vivo paired recordings were not made, presumptive pyramidal cell firing was effectively inhibited by ChR2-expressing SST interneurons activated by short light pulse durations, while pyramidal cell firing returned to baseline upon depolarization block of SST interneurons (**Figure 7B**). The opposite effect was observed in fast-spiking SST cortical interneurons. Increasing light pulse duration onto these interneurons led to enhanced firing (**Figure 7C**) and enhanced inhibition of presumptive cortical pyramidal cells with increasing pulse duration (**Figure 7D**). Verification of ChR2 expression in stimulated neurons in vivo was assumed by evaluating latencies to photo-induced spike activity (median 1.8 ms for presumptive regular-spiking SST interneurons, N = 7, and median 6 ms for presumptive fast-spiking SST interneurons, N = 3). While this is only suggestive that recorded neurons expressed ChR2, whole cell recording data on SST cells that expressed ChR2 revealed that photostimulation results in firing of recorded neurons within a similar range ex vivo (3–8 ms) as observed latencies in vivo. These data are consistent with the assumption that in vivo recordings were made from ChR2-expressing neurons.

Interestingly, mitral cells (**Figure 7E**) displayed enhanced firing rates in response to incrementally increased light pulse durations with no apparent obstruction from depolarization block, though short

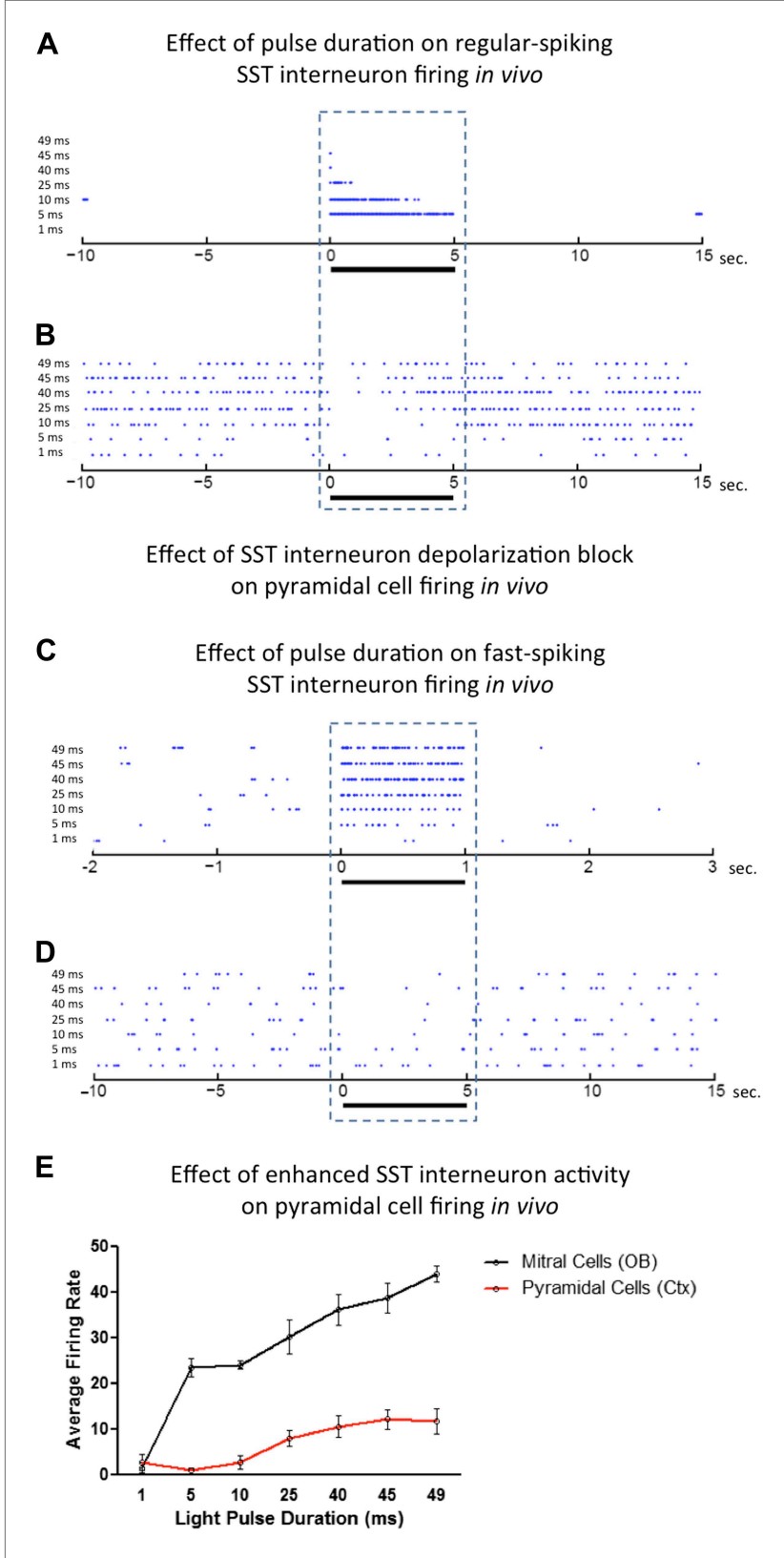

**Figure 7**. Effects of light pulse duration on ChR2-expressing neurons in vivo. Increasing light pulse duration onto (**A**) a presumptive regular-spiking SST cortical interneuron (median latency to spike = 1.8 ms, N = 7 cells from 5 animals) leads to depolarization block and (**B**) disinhibition of a presumptive cortical pyramidal cell. Increasing light pulse

*Figure 7. Continued on next page*

*Figure 7. Continued*

duration onto (**C**) a presumptive fast-spiking SST cortical interneuron (median latency to spike = 6 ms, N = 3 cells from 5 animals) results in enhanced interneuron firing and (**D**) subsequent inhibition of a presumptive cortical pyramidal cell. In contrast to regular-spiking interneurons, increasing light pulse duration onto excitatory (**E**) mitral cells and cortical pyramidal cells enhance average firing rate with increasing pulse width. OB = Olfactory Bulb.

pulse durations were not as effective at eliciting robust neuronal firing as longer pulses, consistent with what we observed previously (*Figure 6E*). However, cortical pyramidal cells (*Figure 7E*) showed lower firing rates than mitral cells, consistent with ex vivo recordings (*Figure 6B*), suggesting that pyramidal neurons exhibit some sensitivity to depolarization block at durations longer than 40 ms, but perhaps not as susceptible as previously noted interneurons.

## Discussion

The use of ChR2 in animal studies is fast becoming a mainstay method to drive neuronal excitability in vivo (*Li et al., 2005*; *Nagel et al., 2005*; *Arenkiel et al., 2007*; *Wang et al., 2007*). Here, we show that targeted expression of ChR2 in specified cell populations of the mammalian brain can lead to depolarization block as a result of prolonged light-induced hyperexcitability. In particular, we demonstrate that the duration of light pulse stimulation used to activate ChR2-expressing neurons is critical to induce consistent and reliable firing of action potentials. Using several mouse lines to drive ChR2 expression in a neuronal subtype-specific manner, we show prominent ex vivo and in vivo silencing of regular-spiking interneuron cell types. In addition, we show that excitatory cell types and fast-spiking interneurons are more resistant to prolonged light-induced depolarization block.

Given its nature as a cation-permeable membrane channel, ChR2-mediated depolarization block is likely due to excessive cation influx into a targeted neuron, resulting in prolonged membrane depolarization. If depolarization block were known to occur in a uniform fashion across interneuron subpopulations, the use of a narrow range of stimulation parameters could reliably avoid spike failures. Currently, however, pulse duration parameters vary widely in the growing number of optogenetic studies, and the use of prolonged light pulses has been employed in optogenetic applications used to drive neuronal dynamics in vivo (*Daou et al., 2013*; *Liske et al., 2013*; *Tabuchi et al., 2013*). Though many optogenetic studies to date have employed short-width pulse parameters for in vivo manipulations, this concern is not inconsequential and should be considered when designing studies that employ ChR2. The physiological relevance of depolarization block is supported by data that show depolarization block can be achieved under normal physiological synaptic function in vivo (*Bianchi et al., 2012*), yet light-induced stimulation of a population of ChR2-expressing neurons is likely to far exceed synaptic activity produced under normal physiological states. Furthermore, and consistent with our data from SST-expressing interneurons, specific populations of seemingly homogeneous neurons in the same brain region may exhibit dramatically different electrophysiological properties, such that some percentage of those neurons exhibit a greater tendency to enter into depolarization block more readily than others (*Unal et al., 2012*). Furthermore, specific biophysical properties of neurons, such as the distribution and number of voltage-gated sodium channels, will influence how susceptible a given class of neurons is to depolarization block to excessive photostimulation (*Tucker et al., 2012*). Similarly, heterogeneity of transgene expression across different driver lines presents another dimension in which cells that express ChR2 may be variably susceptible to block. Likewise, viral-mediated expression of ChR2 with the use of variable-strength promoters is often a more potent method to strongly express ChR2 in cells of interest compared to transgenic models, but this too will lead to further variability. To reduce variability in photo-induced firing responses, it is important to empirically define for each neuronal subset the stimulation parameters that reliably and reversibly activate neurons of interest, before adopting stimulation parameters for in vivo testing.

In vivo optogenetic studies in awake, behaving animals commonly use light stimulation to elicit and measure behavioral or complex physiological changes rather than a change in membrane potential or firing rate of the manipulated neuron (*Adamantidis et al., 2007*; *Aponte et al., 2011*; *Liu et al., 2012*; *Shabel et al., 2012*; *Tan et al., 2012*; *van Zessen et al., 2012*). The facile use of this technology in these contexts lends itself to confound when the appropriate light stimulation parameters are not employed. This stands in contrast to in vivo optophysiology in anesthetized animals, or ex vivo recordings in tissue

slices where firing properties of the cells of interest are directly recorded in response to light exposure. However, when even these experimental preparations are used to study the responses of postsynaptic partners of stimulated neurons, the same concerns apply. In either case, interpretation of the output measure is susceptible to corruption when the cause of changes observed can be attributed to the block of activity in the neuron of interest rather than its excitation. Therefore, it is important to optimize stimulation conditions for targeted activation of a population of neurons. For experiments that require prolonged neuronal activation, the use of step-function opsins (*Berndt et al., 2009*; *Diester and et al, 2011*) might be warranted.

Interestingly, these concerns might also be used advantageously. Consistent with our data, it may be possible to design strategies using ChR2-mediated optical stimulation for the explicit study of depolarization block in which it is thought to subserve a number of neurophysiological processes. Recent studies have highlighted a role for depolarization block in complex neuronal activity (*Marder et al., 1996*; *Grace et al., 1997*; *McIntyre et al., 2004*; *Ullah and Schiff, 2010*; *Bianchi et al., 2012*). For example, states of depolarization block may be a significant factor for information processing in certain classes of neurons (*Marder et al., 1996*; *Dovzhenok and Kuznetsov, 2012*). Interestingly, a long-standing hypothesis regarding the therapeutic action of antipsychotic drugs commonly used in the treatment of schizophrenia features depolarization block in subsets of dopamine neurons after long-term drug treatment (*Grace et al., 1997*; *Boye and Rompre, 2000*; *Valenti et al., 2011*). Similarly, depolarization block has been proposed as a mechanism to explain the therapeutic benefit of deep brain stimulation (*McIntyre et al., 2004*), a method used in the treatment of a variety of movement disorders such as Parkinson's disease. Moreover, persistent sodium currents and consequent depolarization block are thought to facilitate the generation of electrographic seizures (*Bikson et al., 2003*; *Ziburkus et al., 2006*; *Ullah and Schiff, 2010*). With a precise model to control neuronal excitability, or to purposefully induce depolarization block, these phenomena may be topics for future investigation.

Optogenetics affords the ability to mark, map, and manipulate brain cells and circuits with previously unimaginable power and precision. Although the technology to probe brain circuits is rapidly evolving at a breakneck pace, a detailed understanding of the cells being targeted for optogenetic studies remains limited. Our data highlight the need to empirically determine the optimal photostimulation parameters best suited for the cell types being investigated, since as a field we are still learning the possibilities and limitations of optogenetic manipulations.

## Materials and methods

### Experimental mouse lines

Animals were treated in compliance with the US Department of Health and Human Services and Baylor College of Medicine IUCAC guidelines. *Chat-ChR2* (*Zhao et al., 2011*) and *Thy1-ChR2* mice (*Arenkiel et al., 2007*; *Wang et al., 2007*) have been previously described. *Crh-Cre$^{+/−}$* (Crh$^{tm1(cre)Zjh}$) (*Taniguchi et al., 2011*) and floxed conditional ROSA26 ChR2-EYFP female mice (Gt(ROSA)26Sor$^{tm32.1(CAG-COP4*H134R/EYFP)Hze/J}$) were obtained from Jackson Laboratories. *Crh-Cre$^{+/−}$; ROSA26$^{LSL-ChR2-EYFP}$* mice were generated by crossing *Crh-Cre$^{+/−}$* male mice with homozygous floxed conditional ROSA26$^{LSL-ChR2-EYFP}$ female mice. *Sst-Cre$^{+/−}$;* ROSA26$^{LSL-ChR2-EYFP}$ mice were generated by crossing male *Sst-Cre$^{+/−}$* (Sst$^{tm2.1(cre)Zjh/J}$) mice with conditional ROSA26$^{LSL-ChR2-EYFP}$ female mice. *Sst-Cre$^{+/−}$;* ROSA26$^{LSL-tdTomato}$ animals were generated by crossing male *Sst-Cre$^{+/−}$* mice with conditional *ROSA26$^{LSL-tdTomato}$* (B6.Cg-Gt(ROSA)26Sor$^{tm14(CAG-tdTomato)Hze/J}$) female mice.

### Microscopy

Animals were deeply anesthetized using isoflurane and were transcardially perfused with PBS followed by 4% paraformaldehyde (PFA). Brains were dissected and postfixed in 4% PFA for 1 hr at room temperature or overnight at 4°C. Brains were coronally sectioned at 50 µm (olfactory bulb) or 100 µm (forebrain) using a Compresstome (Precisionary Instruments, San Jose, CA). Slices were mounted with Vectashield mounting medium (Vector Laboratories, Burlingame, CA) and detection of EYFP or tdTomato expression was performed using a Leica M205-FA for low-magnification images, and a Leica TCS SPE confocal microscope under a 20X objective for higher magnification images.

### Acute brain slice preparation and electrophysiology

Coronal brain slices (300 µm) were prepared from 3- to 6-week-old animals for all genotypes tested. The slices were embedded in low melting point agarose and sectioned into ice-cold oxygenated (5%

$CO_2$, 95% $O_2$) dissection buffer (in mM: 87 NaCl, 2.5 KCl, 1.6 $NaH_2PO_4$, 25 $NaHCO_3$, 75 sucrose, 10 glucose, 1.3 ascorbic acid, 0.5 $CaCl_2$, 7 $MgCl_2$), recovered for 15 min at 37°C in oxygenated artificial cerebrospinal fluid (ACSF) (in mM: 122 NaCl, 3 KCl, 1.2 $NaH_2PO_4$, 26 $NaHCO_3$, 20 glucose, 2 $CaCl_2$, 1 $MgCl_2$, 305-310 mOsm, pH 7.3), and acclimated at room temperature for 10 min before performing electrophysiological recordings. Borosilicate glass electrodes (Sutter Instruments, Novato, CA) were used for whole cell patch clamp recordings. Electrodes were pulled with tip resistance between 3–8 MΩ, and filled with internal solution (in mM, 120 K-gluconate, 5 KCl, 2 $MgCl_2$, 0.05 EGTA, 10 HEPES, 2 Mg-ATP, 0.4 Mg-GTP, 10 creatine phosphate, 290–300 mOsm, pH 7.3). During recordings, coronal brain slices were placed in a room temperature chamber mounted on an Olympus upright microscope (BX50WI) and perfused with oxygenated ACSF. Cells were visualized under differential interference contrast imaging. Data were obtained via a Multiclamp 700B amplifier, low-pass Bessel-filtered at 4 kHz, and digitized on computer disk (Clampex, Axon Instruments). Excitation light was from a BLM-Series 473 nm blue laser system (Spectra Services, Ontario, NY), which was controlled by digital commands from Clampex to trigger photostimulation. The firing rate for each cell recorded for each stimulation parameter was counted manually for the full stimulus train and then averaged and plotted using GraphPad Prism statistical software (GraphPad Software Inc, La Jolla, CA.). Light-evoked spike amplitudes were measured for the duration of the full stimulus train. Amplitudes were normalized to the initial light-evoked action potential for each respective trace, and temporally corresponding amplitudes were averaged for each cell type for a given pulse duration parameter. Spike probabilities were calculated by counting successful light-evoked spikes for each trace of a corresponding stimulus parameter and averaging success rates across cells for each cell type tested. Minimal pulse widths were determined by photostimulating neurons starting with short (sub-millisecond) to high (1 ms) pulse widths. Criteria for consideration of minimal pulse width was that a given pulse width was capable of eliciting ≥70% fidelity (i.e., a minimum of seven light-evoked action potentials out of 10 pulses). Minimal pulse widths were then averaged across cells for each cell type tested.

## In vivo photostimulation and electrophysiology

For in vivo recordings, animals were injected IP with ketamine (150 µg/g body weight), followed by sustained delivery of 0.3% isoflurane with oxygen to the animal. For olfactory bulb recordings, the dorsal surface of the olfactory bulb was carefully exposed so as to not damage the pia or underlying brain tissue. For extracellular recordings from the cortex, a small area of the skull overlying the cortical region of interest was removed to expose the underlying brain tissue. For light stimulation and electrophysiological recordings, optrodes made from fiber optics and 1.0 MΩ extracellular tungsten recording electrodes (Microprobe Inc., Gaithersburg, MD) were used. A blue laser source (CrystaLaser, Reno, NV) was controlled by a Master-8 (AMPI, Israel), and guided to either the olfactory bulb or cortex by focusing light onto fused silica fiber optics. Extracellular recordings were amplified by a Model 1800 AC amplifier (A-M systems, Carlsborg, WA), digitized using a CED Power 1401 mk II (Cambridge Electronic Design, Cambridge, England), and processed using Spike2 acquisition software (Cambridge Electronic Design, Cambridge, England). For in vivo recordings, ChR2 expression was suggested by short latency to spike. Median latencies were calculated using MATLAB software. In addition to latencies, mitral cell identity can be determined by characteristic firing that is coupled to respiration. Thus, ChR2 expression can be indirectly determined by monitoring changes in respiratory-coupled firing responses when a ChR2-expressing cell is activated by light.

## Additional information

### Funding

| Funder | Grant reference number | Author |
| --- | --- | --- |
| National Institute of Neurological Disorders and Stroke, National Institutes of Health | R01NS078294 | Alexander M Herman, Dona K Murphey, Isabella Garcia, Benjamin R Arenkiel |
| McNair Medical Institute | | Longwen Huang, Benjamin R Arenkiel |

| Funder | Grant reference number | Author |
|--------|------------------------|--------|
| National Institute of Neurological Disorders and Stroke, National Institutes of Health | F31NS081805 | Isabella Garcia |
| National Institute of Neurological Disorders and Stroke, National Institutes of Health | R00NS064171 | Benjamin R Arenkiel |

The funders had no role in study design, data collection and interpretation, or the decision to submit the work for publication.

## Author contributions

AMH, DKM, IG, Acquisition of data, Analysis and interpretation of data, Drafting or revising the article; LH, Acquisition of data, Analysis and interpretation of data; BRA, Conception and design, Analysis and interpretation of data, Drafting or revising the article

## Ethics

Animal experimentation: This study was performed in strict accordance with the recommendations in the Guide for the Care and Use of Laboratory Animals of the National Institutes of Health. All of the animals were handled according to approved institutional animal care and use committee (IACUC) protocols (AN5596) of Baylor College of Medicine. All surgical procedures were performed under anesthesia, and every effort was made to minimize pain.

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
