## [Decision Letter]

Thank you for sending your work entitled “Cell type-specific and time-dependent light exposure contribute to silencing in neurons expressing Channelrhodopsin-2” for consideration at *eLif*e. Your article has been favorably evaluated by a Senior editor, a Reviewing editor, and 2 reviewers.

The following individuals responsible for the peer review of your submission have agreed to reveal their identity: Ronald L Calabrese (Reviewing editor) and Antoine Adamantidis (Reviewer 1).

The Reviewing editor and the two reviewers discussed their comments before we reached this decision, and the Reviewing editor has assembled the following comments to help you prepare a revised submission.

In this study, Herman et al. systematically characterize the fidelity of optogenetically-induced spiking in inhibitory and principal neurons from the mouse brain using in vitro and in vivo electrophysiology. This timely technical report emphasizes the importance of empirical characterization of cell response to optogenetic stimulations. The authors show that inhibitory interneurons, as well as some pyramidal cells may enter in a depolarization block upon prolonged continuous optical stimulation. The work shows that such validation is crucial to determine that cells are actually activated and not inhibited during optogenetic stimulation. While depolarization block by long-duration optical pulses had already been observed, it is very useful to have this systematic assessment of the phenomenon and to demonstrate that susceptibility to depolarization block can vary across cell type.

While the reviewers thought the work very worthwhile and interesting there were some concerns that should be addressed. The reviews are highly congruent and complementary, and they are provided below so that the authors can benefit from them directly. The authors should address each concern of both reviewers. Major emphasis in the revision should focus on:

1) The authors should cite other relevant reports of optogenetically induced depolarization block as detailed in the reviews.

2) The authors should explore how the degree of depolarization block depends on optical stimulation intensity; i.e., show at least some data on the effect of changing light intensities (starting at low power levels) on depolarization block as detailed in the reviews.

3) The authors should carefully report quantitative data as detailed in the reviews.

4) In the in vivo experiments, the authors should address the issue of whether the recorded neuron expresses ChR2 as detailed in the reviews.

5) The authors should consider the impact of pulse duration in optogenetic stimulation and depolarization block as detailed in the reviews.

Reviewer #1 summary and major comments:

The study by Herman and collaborators aims at characterizing the fidelity of optogenetically-induced spike in inhibitory and principal neurons from the mouse brain using in vitro and in vivo electrophysiology.

This timely technical report emphasizes the importance of empirical characterization of cell response to optogenetic stimulations, which is not trivial and important for the field. Importantly, the authors show that inhibitory interneurons, as well as some pyramidal cells may enter in a depolarization block upon prolonged continuous optical stimulation. As noted by the authors, this validation is crucial to show that cells are actually activated and not inhibited during optogenetic experiments.

The manuscript is very clear, well organized and extremely well written. My major comments relate to the graphical representations in the figures and the data analysis.

1) Graphical representations of the firing response refer to the “average firing rate” (Figures 4 and 5): although it is a useful quantification, it is a cell-dependent phenomena that varies a lot amongst neuronal population. Representing the fidelity of light-evoked spikes (over the entire stimulation or the first 50 spikes, for instance) is more accurate and should be reported.

2) Many cell types show spike doublets/triplets upon light stimulation. This should be better characterized and quantified. Similarly, it would be interesting and useful for optogenetic users to report the minimal light pulse width (1 ms or below?) necessary to evoke single spike for the different cell types tested in this study.

3) The in vivo recording experiment is interesting per se. However, it brings an important question: Does the recorded cell express ChR2, or does the recorded spike result from an indirect activation or disinhibition? One way to address this is to measure the latency to spike. However, a short latency to spike only suggests, but not demonstrates, that the recorded cell is expressing ChR2. To my knowledge, only juxtacellular techniques can definitely prove that the recorded cell is positive for ChR2. Thus latencies to spike should be reported in the figures since those might also differ between cells and this not trivial point should be discussed in the manuscript.

4) The manuscript would benefit from a brief summary and discussion of the possible mechanism of the depolarization block.

Reviewer #2 summary and major comments:

This manuscript demonstrates that high-duty-cycle optical stimulation of channelrhodopsin2 (ChR2-H134R) can lead to a suppression of spiking due to depolarization block in many, but all, cell types. This is demonstrated in slices and with in vivo recordings. While depolarization block by long-duration optical pulses has certainly been noted (see comments below), it is useful too have this systematic assay of the phenomenon and to demonstrate that the susceptibility to depolarization block can vary across cell type. The manuscript would be improved by addressing several issues.

1) The basic phenomenon of depolarization block has been widely reported in a number of papers characterizing channelrhodopsin behavior. To take one recent example, Lin et al., Nat Neurosci 16:1499-508 compared depolarization block due to activation of a number of different ChR2 variants (e.g., Figure 3). In general, the authors should make this more clear in the Introduction and/or Discussion.

2) One would expect the degree of depolarization block to also depend on optical stimulation intensity. In these experiments all stimulation was performed with high light intensities (40 mW/mm^2^). It would be useful to show at least some examples of the effect of changing light intensities (starting at much lower power levels) on depolarization block.

3) In general, quantitative results are not reported carefully. For example there are no error bars on the plots in Figure 1 and Figure 4, nor are error bars in Figure 5 defined. It would also be useful to report number of animals (not just cells) tested for each dataset. In Panels 6E and F, there are bar plots with error bars, but it is not stated what the error bars represent nor what the values represent: how many neurons were recorded, from how many preparations? For that matter it seems that the data in 6E and F are taken from different animals than in panels A–D, but this also is not stated.

4) While it is clearly useful to demonstrate that depolarization block can occur with long optical pulses, in this reviewer's experience the large majority of published papers use pulses of shorter duration in which this is not a major consideration. In the Discussion, the authors state that it is ‘not uncommon’ for studies to use longer, whole-second pulses of light for ChR2 activation. However, no studies are cited. It is important to cite some specific examples.

---

## [Author Response]

*Reviewer #1 summary and major comments*:

*1) Graphical representations of the firing response refer to the “average firing rate” (*Figures 4 and 5*): although it is a useful quantification, it is a cell-dependent phenomena that varies a lot amongst neuronal population. Representing the fidelity of light-evoked spikes (over the entire stimulation or the first 50 spikes, for instance) is more accurate and should be reported*.

To address the issue of spike fidelity in response to differing light pulse durations, we have further measured and reported both ‘spike probabilities’ and ‘average change in amplitudes’ over the entire stimulation train for each cell type tested.

*2) Many cell types show spike doublets/triplets upon light stimulation. This should be better characterized and quantified. Similarly, it would be interesting and useful for optogenetic users to report the minimal light pulse width (1 ms or below?) necessary to evoke single spike for the different cell types tested in this study*.

To better characterize the firing responses from tested ChR2-expressing cells, we have measured and provided additional data for the minimal pulse widths required to elicit single action potentials from each of the cell types tested.

*3) The in vivo recording experiment is interesting per se. However, it brings an important question: Does the recorded cell express ChR2, or does the recorded spike result from an indirect activation or disinhibition? One way to address this is to measure the latency to spike. However, a short latency to spike only suggests, but not demonstrates, that the recorded cell is expressing ChR2. To my knowledge, only juxtacellular techniques can definitely prove that the recorded cell is positive for ChR2. Thus latencies to spike should be reported in the figures since those might also differ between cells and this not trivial point should be discussed in the manuscript*.

To better show that in vivo recordings were made from ChR2-expressing neurons, we have quantified and discussed the latency to spike for our in vivo experiments.

*4) The manuscript would benefit from a brief summary and discussion of the possible mechanism of the depolarization block*.

We have elaborated in the Discussion section possible mechanisms of light-induced depolarization block in neurons expressing ChR2, as well as providing additional references related to the topic.

*Reviewer #2 summary and major comments*:

*1) The basic phenomenon of depolarization block has been widely reported in a number of papers characterizing channelrhodopsin behavior. To take one recent example, Lin et al., Nat Neurosci 16:1499-508 compared depolarization block due to activation of a number of different ChR2 variants (e.g.,*
Figure 3*). In general, the authors should make this more clear in the Introduction and/or Discussion*.

In our Introduction, we have more thoroughly discussed relevant findings from previous publications.

*2) One would expect the degree of depolarization block to also depend on optical stimulation intensity. In these experiments all stimulation was performed with high light intensities (40 mW/mm*^*2*^*). It would be useful to show at least some examples of the effect of changing light intensities (starting at much lower power levels) on depolarization block*.

To address the impact of light intensity on depolarization block, we have performed additional experiments where we recorded and reported average firing rates for all cell types tested in response to different optical intensities (≈13 mw/mm^2^, 20 mw/mm^2^, 26 mw/mm^2^, 40 mw/mm^2^).

*3) In general, quantitative results are not reported carefully. For example there are no error bars on the plots in*
Figure 1
*and*
Figure 4*, nor are error bars in*
Figure 5
*defined. It would also be useful to report number of animals (not just cells) tested for each dataset. In Panels 6E and F, there are bar plots with error bars, but it is not stated what the error bars represent nor what the values represent: how many neurons were recorded, from how many preparations? For that matter it seems that the data in 6E and F are taken from different animals than in panels A*–*D, but this also is not stated*.

To address these concerns, we have included and defined error bars in appropriate figures, and we have elaborated in the text the number of animals, cells, and preparations used for recordings. We have also elaborated within the ‘Material and methods’ section on the statistical analyses performed.

*4) While it is clearly useful to demonstrate that depolarization block can occur with long optical pulses, in this reviewer's experience the large majority of published papers use pulses of shorter duration in which this is not a major consideration. In the Discussion, the authors state that it is ‘not uncommon’ for studies to use longer, whole-second pulses of light for ChR2 activation. However, no studies are cited. It is important to cite some specific examples*.

To clarify our Discussion, we have included several relevant citations in which continuous light pulses have been used for in vivo optogenetic experiments. In addition, we provide a more comprehensive discussion of the phenomenon of depolarization block with respect to optogenetic manipulations.